# Diet-Wide Association Study for the Incidence of Type 2 Diabetes Mellitus in Community-Dwelling Adults Using the UK Biobank Data

**DOI:** 10.3390/nu16010103

**Published:** 2023-12-28

**Authors:** Jiahao Liu, Xianwen Shang, Yutong Chen, Wentao Tang, Mayinuer Yusufu, Ziqi Chen, Ruiye Chen, Wenyi Hu, Catherine Jan, Li Li, Mingguang He, Zhuoting Zhu, Lei Zhang

**Affiliations:** 1Faculty of Medicine, Dentistry and Health Sciences, University of Melbourne, Melbourne, VIC 3010, Australia; jiahaol7@student.unimelb.edu.au (J.L.); myusufu@student.unimelb.edu.au (M.Y.); ziqic3@student.unimelb.edu.au (Z.C.); ruiyec@student.unimelb.edu.au (R.C.); wenyih1@student.unimelb.edu.au (W.H.); catherine.jan@student.unimelb.edu.au (C.J.); lili7@student.unimelb.edu.au (L.L.); 2Centre for Eye Research Australia, Royal Victorian Eye and Ear Hospital, Melbourne, VIC 3002, Australia; xianwen.shang@unimelb.edu.au (X.S.); mingguang.he@unimelb.edu.au (M.H.); lisa.zhu@unimelb.edu.au (Z.Z.); 3Guangdong Eye Institute, Department of Ophthalmology, Guangdong Provincial People’s Hospital, Guangdong Academy of Medical Sciences, Guangzhou 510080, China; 4Faculty of Medicine, Nursing and Health Science, Monash University, Clayton, VIC 3800, Australia; yche0446@student.monash.edu; 5Faculty of Medicine and Health, University of Sydney, Sydney, NSW 2006, Australia; wentaot@student.unimelb.edu.au; 6Centre for Epidemiology and Biostatistics, Melbourne School of Population and Global Health, University of Melbourne, Melbourne, VIC 3053, Australia; 7Ophthalmology, Department of Surgery, University of Melbourne, Melbourne, VIC 3052, Australia; 8State Key Laboratory of Ophthalmology, Zhongshan Ophthalmic Center, Sun Yat-sen University, Guangzhou 510060, China; 9Department of Nephrology, State Key Laboratory of Reproductive Medicine, Children’s Hospital of Nanjing Medical University, Nanjing 210008, China; 10Melbourne Sexual Health Centre, Alfred Health, Melbourne, VIC 3053, Australia; 11Central Clinical School, Faculty of Medicine, Monash University, Melbourne, VIC 3168, Australia

**Keywords:** type 2 diabetes mellitus, diet-wide association study, diet, nutrition, nutrient

## Abstract

This longitudinal study used diet-wide association studies (DWAS) to investigate the association between diverse dietary food and nutrient intakes and the onset of type 2 diabetes mellitus (T2DM). Out of 502,505 participants from the UK Biobank, 119,040 with dietary data free of T2DM at the baseline were included, and 3241 developed T2DM during a median follow-up of 11.7 years. The DWAS analysis, which is based on Cox regression models, was used to analyse the associations between dietary food or nutrient intake factors and T2DM risk. The study found that 10 out of 225 dietary factors were significantly associated with the T2DM risk. Total alcohol (HR = 0.86, 0.85–0.92, *p* = 1.26 × 10^−32^), red wine (HR = 0.89, 0.88–0.94, *p* = 7.95 × 10^−19^), and fresh tomatoes (HR = 0.92, 0.89–0.94, *p* = 2.3 × 10^−11^) showed a negative association with T2DM risk, whereas sliced buttered bread exhibited a positive association. Additionally, 5 out of 21 nutrient intake variables revealed significant associations with the T2DM risk, with iron having the highest protective effect and starch as a risk factor. In conclusion, DWAS is an effective method for discovering novel associations when exploring numerous dietary variables simultaneously and could provide valuable insight into future dietary guidance for T2DM.

## 1. Introduction

Type 2 diabetes mellitus (T2DM) is a common metabolic disease that has been labelled as a ‘21st Century epidemic’ [1]. According to the International Diabetes Federation, the prevalence of diabetes among adults aged 20–79 years worldwide has reached 10.5% (536.6 million) in 2021, and it is expected to reach 12.2% (643 million) in 2045 [2,3]. Furthermore, T2DM is responsible for over 1.5 million deaths each year [4]. Given these alarming statistics, it is crucial to identify modifiable risk factors for T2DM to enhance current prevention and self-management strategies.

Dietary modification plays a crucial role in preventing and managing T2DM. Studies have identified certain dietary risk factors associated with T2DM, such as saturated fatty acids and low-calorie ketogenic diet patterns [5,6]. Previous research has also suggested that higher adherence to a ‘Mediterranean diet pattern’ can benefit T2DM prevention [7]. However, these studies only addressed the contributable effects of specific dietary pattern subcategories, which limited our understanding of the impact of individual nutrients on T2DM risk. Very few dietary studies have comprehensively examined a vast number of dietary factors simultaneously. Furthermore, while previous studies have identified key pathways contributing to T2DM onset [8], the specific biological and physiological mechanisms that underlie the association between dietary food intake, nutrient intake, and T2DM are not fully understood.

Recently, a novel method called diet-wide association studies (DWAS) has been developed, which can investigate associations between dietary food and nutrient intake and disease phenotypes [9]. This method was inspired by existing Genome-wide association studies (GWAS) and phenome-wide association studies (PhenWAS), which examine how different exposure levels of risk factors can lead to changes in the phenotype [9,10]. The GWAS approach has been extensively utilized to establish the connections between diseases and single-nucleotide polymorphisms (SNPs) [9]. In contrast, PhenWAS is a method that focuses on traits or phenotypes by integrating a specific metabolic or environmental exposure factor. A prior study utilizing the PhenWAS approach demonstrated that higher circulating lipoprotein levels were associated with cardiac diseases, genitourinary system diseases, and T2DM [10]. DWAS is an effective method for discovering novel associations when exploring a vast number of dietary variables simultaneously and may provide valuable insight to guide the early prevention of T2DM.

This study aims to employ the DWAS approach to identify protective and risk factors of T2DM-related nutrients and dietary food intakes, using the comprehensive population-based nutrition data provided by the UK Biobank.

## 2. Method

For this study, data were sourced from the UK Biobank population-based cohort. Between 2006 and 2010, over 500,000 individuals between the ages of 40 and 69 were recruited from the United Kingdom [11]. Participants were asked to complete touch-screen self-report questionnaires, online computer-assisted personal interviews, or attend one of the 22 specified assessment centres located in England, Scotland, and Wales for physical measurements [11].

### 2.1. Ascertainment of T2DM

Baseline diagnosis criteria were utilized through hospital inpatient records, death registry, and self-report data. For hospital admission data, T2DM identification was based on International Classification of Diseases (ICD) codes, including 25001, 25003, 25011, 25013, 25021, 25023, 25031, 25033, 25041, 25043, 25051, 25053, 25061, 25063, 25071, 25073, 25081, 25083, 25091, 25093 in ICD-9 and E10, E11, E13, and E14 in ICD-10 [12]. Participants who had previously received a T2DM diagnosis before the dietary survey assessment were excluded to prevent reverse causality bias. Incident T2DM was determined using hospital inpatient records and death registry until January 2021 (E10, E11, E13, and E14 in ICD-10).

### 2.2. Dietary and Nutrient Intake

The evaluation of dietary survey assessments involved the use of an online 24 h recall self-administered questionnaire (EPIC-soft 24 h recall questionnaire) on one or more of the five occasions between April 2009 and June 2012 [11]. The average of the number of assessment questionnaires was utilized to calculate dietary intakes, and the information was recorded in servings/week or transformed from gram/week to gram/day. The quantity of each type of drink was multiplied by its standard drink size and reference alcohol content, then converted to consumption per day. Nutrient information was calculated using internal algorithms and information from the sixth edition of the Composition of Foods (2002) [11]. Additionally, energy-adjusted dietary food and nutrient intake, apart from alcohol intakes, were computed using the formula of residuals for the subject from a regression model with dietary intake and total energy intake as the independent variable plus the expected dietary intake for a person with mean energy intake [13]. Finally, energy-adjusted dietary intake and nutrient intake factors were divided into quintiles for further analysis.

### 2.3. Covariates

The demographic information that was collected included gender (male or female), age (in years), ethnicity (White, White and White mixed background, Asian and Asian mixed background, or other ethnic background), education level (high, intermediate, or low), and income (categorized as <£18,000, £18,000–£30,999, £31,000–£51,999, £52,000–£100,000, >£100,000, or unknown). Participants self-reported this information. Lifestyle factors, such as physical activity (measured in metabolic equivalents minutes per week), sleep duration (categorized as ≥7 h/day and ≤9 h/day or ≤7 h/day and ≥9 h/day), smoking status (never, former, current, or unknown), and metabolic rate (calculated using the Oxford equation), were also collected [14]. Height and weight were measured using a Tanita BC-418 MA body composition analyser, and body mass index (BMI) was calculated [11]. More information on the study can be found on the UK-Biobank website [15]. The maximum missing rate observed was 15.1% for physical activity data. To address this, missing covariate values were imputed using a multiple imputation method, a five-fold approach with a random forest algorithm.

### 2.4. Statistical Analysis

The baseline characteristics data were presented as frequency (percentage [%]) and mean (standard deviation [SD]). To assess the differences in baseline characteristics between the non-T2DM and T2DM onset groups, we conducted univariate analyses using chi-square tests for categorical covariates and one-way analysis of variance (ANOVA) for continuous covariates. X^2^ tests were conducted for gender, ethnic background, smoking status, education level, medicine for exogenous hormones, blood pressure, and medical condition of hypertension. ANOVA analysis was conducted for age at recruitment, alcohol intake, physical activity, Townsend index, BMI, haemoglobin A1C (HbA1c), and blood cholesterol. We also ensured that the syntax, grammar, choice of words, and use of adjectives and adverbs were correct and appropriate.

### 2.5. DWAS

The study conducted DWAS by using Cox regression models to analyse the pseudo-numerical associations between the quintiles of dietary food or nutrient intake factors for T2DM risk [16]. Model 1 was adjusted for gender and age at recruitment. Model 2 was additionally adjusted for ethnic background, smoking status, physical activity, education level, and Townsend index. Model 3 was further adjusted for BMI, HbA1c, blood cholesterol, medication for blood pressure and exogenous hormone, and medical condition of hypertension. For Model 2 and Model 3, non-alcoholic dietary food and nutrient variables were additionally adjusted for alcohol intake. We also applied the Bonferroni correction method for the Model 1 *p*-value, and the *p*-value threshold was set at 0.05 [17]. To visualize the DWAS result, we introduced Manhattan plots and forest plots. Additionally, we conducted Spearman’s correlation analysis to assess the collinearity among dietary factors, and a heatmap was created to visualize the correlation patterns [18]. Outcomes were validated through 1000-times bootstrap resampling using Model 3.

Two Sensitivity analysis was conducted for the validated significant associations: (1) to re-examine the associations in participants who had three times or more dietary assessment records, (2) to re-examine the associations by excluding participants with nutrient intake among the top and bottom 1%, for dietary food intakes Proportional hazard assumption analysis was performed for significant dietary factors that were recorded in detail and not categorized by measurement units. Kaplan–Meier survival curves visualized the result.

The study utilized STATA version 16.0 (StataCorp LLC, College Station, TX, USA) for data preparation and R version 4.2.3 (The R Foundation for Statistical Computing c/o Institute for Statistics and Mathematics, Vienna, Austria) with RStudio Desktop 4.3.0 for statistical analysis. R packages that have been used for multiple imputations were “mice”, for the baseline characteristics comparison between non-T2DM and T2DM onset individuals was “tableone”, for the NWAS analysis was “survival”, for NWAS visualization was “ggolot”, for correlation analysis was “corrplot”, and for Kaplan–Meier survival curves is “survminer”.

## 3. Results

### 3.1. Population Selection

Out of the total 502,505 participants, we included 119,040 participants in the analysis, comprising 67,364 females (56.6%) and 51,676 males (43.4%) (Figure 1). The mean age of the participants was 56.0 years (SD: 7.9 years), as presented in Table 1.

### 3.2. Incidence of T2DM

Over a follow-up period of 1,398,053 person years (with a median follow-up period of 11.8 years), 3241 new T2DM cases were recorded, as shown in Table 1. The incidence rate of T2DM was found to be 23.2 incident cases per 10^4^ person years (incident/10^4^ py), with a confidence interval [CI] of 22.4–24.0 incident/10^4^ py.

### 3.3. DWAS

Of 224 dietary food intake factors (Appendix A), 10 protective factors and 4 risk factors were found to be significantly associated with the onset risk of T2DM (Figure 2a–c and Figure 3a, Appendix A). Of 21 dietary nutrient intake factors (Appendix A), 1 protective factors and 2 risk factor (starch) were found to be significantly associated with the onset risk of T2DM (Figure 2d–f and Figure 3b, Appendix A). We also reported the results of factors (Hazard ratio [HR], 95% confidence interval [CI], *p*-value [p]) that consistently demonstrated significant association across all models (Appendix A).

#### 3.3.1. Estimated Dietary Foods

For grain, buttered sliced bread (HR = 1.09, 1.06–1.12, *p* = 3.12 × 10^−11^) was the sole risk factor for T2DM risk. For vegetables, fresh tomatoes (HR = 0.92, 0.89–0.94, *p* = 2.3 × 10^−11^) was a protective factor against T2DM onset. Two types of fruit were protective factors against T2DM onset, including apple (HR = 0.95, 0.92–0.97, *p* = 1.17 × 10^−5^) and berries (HR = 0.96, 0.92–0.98, *p* = 0.002). Oily fish (HR = 0.95, 0.92–0.99, *p* = 9.7 × 10^−5^) was a protective factor for T2DM risk. For dairy products, milk (HR = 1.03, 1.01–1.06, *p* = 0.018) was a risk factor for T2DM onset. For non-alcoholic beverages, filtered coffee (HR = 0.93, 0.90–0.96, *p* = 4.62 × 10^−9^) was the sole protective factor. While fruit drinks (HR = 1.03, 1.01–1.07, *p* = 0.007) were a risk factor for T2DM. Three types of alcoholic beverages were protective factors for T2DM risk, including total alcohol (HR = 0.86, 0.85–0.92, *p* = 1.26 × 10^−32^), red wine (HR = 0.89, 0.85–0.92, *p* = 7.95 × 10^−19^), and white wine (HR = 0.91, 0.89–0.96, *p* = 1.31 × 10^−12^) (Figure 3a). Lettuce (HR = 0.97, 0.91–0.99, *p*-value = 0.006) and mixed side salad (HR = 0.96, 0.93–1.00, *p*-value = 0.003) also exhibited significant associations with T2DM risk, albeit their effects were marginal.

#### 3.3.2. Estimated Dietary Nutrients

For nutrient factors, iron (HR = 0.95, 0.92–0.98, *p* = 2.43 × 10^−4^) was a protective factor for T2DM risk. While starch (HR = 1.06, 1.01–1.09, *p* = 1.22 × 10^−5^) was a nutrient risk factor for T2DM risk (Figure 3b). Magnesium (HR = 0.96, 0.93–1.00, *p*-value = 0.005), potassium (HR = 0.96, 0.93–1.00, *p*-value = 0.005) and vitamin E (HR = 0.96, 0.94–1.00, *p*-value = 0.003) showed a slight protective effect that has been used for multiple imputations, although the magnitude of these associations was modest.

### 3.4. Sensitivity Analysis

In the sensitivity analyses, sliced buttered bread, fresh tomatoes, apples, berries, oily fish, filtered coffee, total alcohol, red wine, and white wine consumption remained significantly associated with T2DM risk. For nutrient intakes, iron and starch remained significantly associated with T2DM risk for nutrient intakes in both sensitivity analyses. All proportional hazard assumptions were satisfactorily validated, except iron intake, which demonstrated a violation of the assumption.

### 3.5. Dietary Factors Correlations

The majority of correlations between dietary factors were found to be non-significant (*p* > 0.05), except for positive correlations between oily fish and milk (correlation = 0.42), milk and fruit drinks (correlation = 0.4), and total alcohol and red wine (correlation = 0.47), and a negative correlation between berries and fresh milk (correlation = −0.36) (Figure 4).

## 4. Discussion

This study represents the first DWAS investigation to simultaneously examine the impact of a wide range of dietary and nutrient intake factors on T2DM risk. We analysed 224 dietary and 21 nutrient intake factors, of which 9 dietary foods and 1 nutrient (iron) showed a protective effect against the onset of T2DM, while 1 dietary food (sliced buttered bread) and 1 nutrient (starch) were identified as risk factors. Furthermore, our study revealed that total alcohol, white wine, red wine, and fresh tomato intake exhibited the most notable protective effects against T2DM, whereas the consumption of sliced buttered bread was associated with an increased risk of T2DM.

Our study has shown that most plant-based foods protect against the onset of T2DM, especially fresh tomatoes, apples, and berries. While sliced buttered bread showed a detrimental effect on T2DM risk. These findings align with a previous meta-analysis, which reported a 23% reduction in T2DM onset with a high intake of plant-based dietary patterns [19]. The beneficial effects can be attributed to their bioactive compounds, including plant-based fibre, polyphenols, antioxidants, and flavonoids, which combat LDL oxidation, reduce circulating C-reactive protein levels, increase circulating carotenoid levels, improve insulin sensitivity, reduce inflammation, and modify the composition of the gut microbiota [19,20]. Moreover, a previous study highlighted the potential benefits of fresh tomatoes in the aetiology of T2DM, owing to the antioxidative and anti-inflammatory effects of lycopene in tomatoes [21]. However, our results for fruit juice exhibited a detrimental effect consistent with a previous meta-analysis that revealed a 6% increment in the risk of T2DM [22]. For fresh tomato intake, our findings align with those from a randomized controlled trial that reported a 3-fold increase in lycopene concentration in the LDL fractions of T2DM subjects who consumed tomato juice compared to the control group [21]. Additionally, mixed side salad exhibited a marginally protective effect, highlighting the importance of the glycaemic level of different plant-based foods and the protective effect of plant-based fibre. Regarding sliced buttered bread intake, it is conceivable that the protective effect of plant-based foods in grain intake could be masked by the presence of high-fat content in plant-based foods [23].

According to our study, consuming oily fish such as salmon, tinned salmon, herring, mackerel, sardines, and fresh tuna steak has been a protective factor for T2DM. The long-chain omega-3 polyunsaturated fatty acids like eicosapentaenoic acid and docosahexaenoic acid are the key factors contributing to lowering plasma triglycerides and oxidative stress [24]. Our result is consistent with the previous UK Biobank study that participants who consumed oily fish reduced T2DM risk by 9%; the mitigated protective effect could be attributed to the excluding participants who had cardiovascular disease and cancer at the baseline [25].

Beverages such as filtered coffee and alcoholic beverages, including total alcohol, red wine, and white wine intakes, have been shown to be protective against the risk of T2DM. Filter coffee has been shown in previous studies to have protective effects both phenotypically and genetically by improving glucose metabolism and reducing BMI [26]. However, it is important to note that decaffeinated and instant coffee have shown detrimental effects [26], and further research is needed to unravel the underlying pathophysiological mechanisms. Additionally, our finding on the preventive effect of alcohol and wine consumption aligns with a previous longitudinal study [27]. The presence of natural phytochemical compounds, such as polyphenols with antioxidant properties, has been demonstrated to improve insulin sensitivity [27].

For dietary nutrients, starch is the sole risk factor, which has been found to enhance the risk of T2DM, whereas iron was the sole protective factor against T2DM. For starch intake, our results were consistent with a study conducted on US females, which reported that increased starch intake is linked to a 23% higher T2DM risk [22]. Although the effect size observed in our study may differ from the previous study, this discrepancy could be attributed to differences in study populations. Our study included both male and female participants. Nevertheless, the evidence indicates that starch intake significantly increases the risk of developing T2DM. Our findings on iron consumption diverged from a previous study, which identified a positive association between heme-iron and T2DM risk [28]. This discrepancy may stem from the violation of the proportional hazard assumption in our analysis. Given that heme-iron predominantly originates from meat sources, it is crucial to consider the balancing interplay between heme-iron from meat and non-heme iron from plant-based sources. Further research is warranted to elucidate the true relationship between iron consumption and T2DM risk.

There are several limitations to this study that should be noted. Firstly, it is possible that a reverse causal effect may apply to participants with pre-existing illnesses, obesity, or a family history of T2DM, as changes in their eating behaviour may have occurred prior to the study. Secondly, the study did not consider the non-linear association between individual dietary factors and T2DM risk. Thirdly, the analysis treated each factor independently, which means that collinearity between different dietary factors could not be addressed. Fourthly, all dietary information was self-reported, which may be subject to recall bias. Finally, the study’s generalizability to other ethnicities and nationalities may be limited, as over 97.5% of the data come from individuals with a White Caucasian background from the UK, and the proportion of females in this study (56.6%) is slightly higher than the proportion in the UK general population (51.0%).

## 5. Conclusions

The study found that out of 224 dietary food variables and 21 nutrient intake variables, 23 and 7, respectively, were significantly associated with the onset of T2DM. The analysis revealed that white wine, red wine, and fresh tomatoes had protective effects, while sliced buttered bread and low-calorie drinks posed a risk for T2DM. This study also identified vitamin E, potassium, and iron as protective factors among dietary nutrients. This article not only provides valuable insights into future dietary guidance for T2DM prevention and management strategies but also paves the way for further research into the crucial role of nutritional factors in preventing T2DM.

## Figures and Tables

**Figure 1 nutrients-16-00103-f001:**
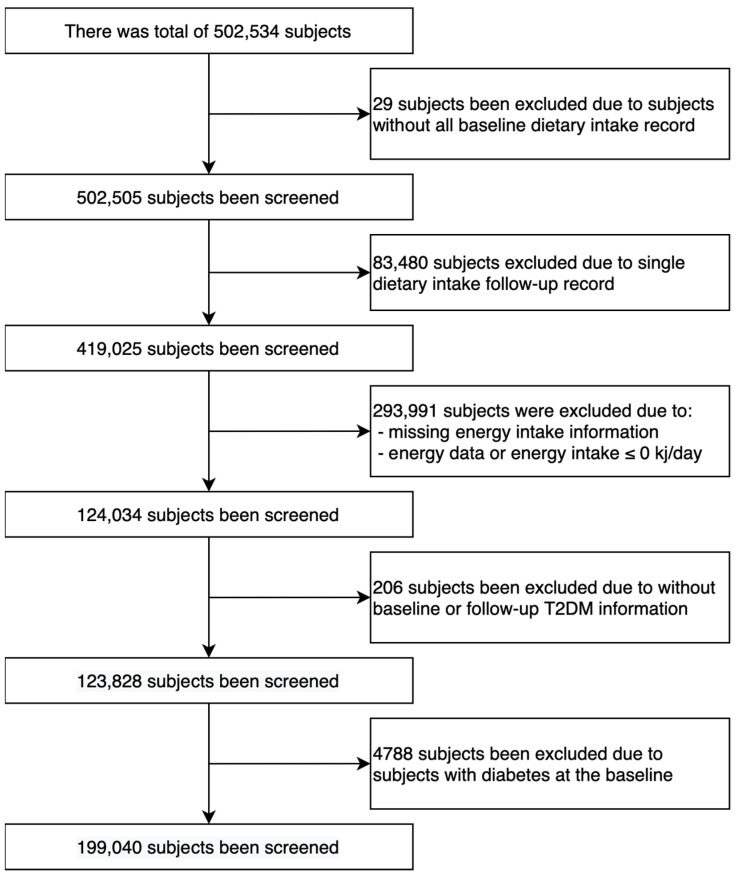
The flowchart of sample selection.

**Figure 2 nutrients-16-00103-f002:**
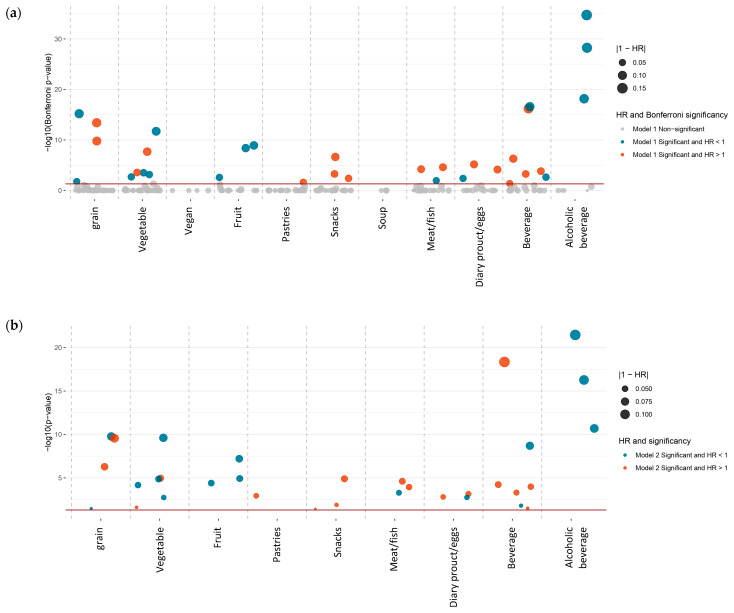
Manhattan plot of the diet-wide association study (DWAS) for the T2DM risk (**a**–**c**) for dietary food factors and (**d**–**f**) for dietary nutrient factors. Abbreviation: HR, hazard ratio. The HR represents the increased T2DM risk for each quintile increase in dietary factor consumption. (**a**,**d**) Model 1 is adjusted for gender and age at recruitment. (**b**,**e**) Model 2 is adjusted for ethnic background, smoking status, physical activity, education level, and Townsend index. (**c**,**f**) Model 3 is further adjusted for BMI, HbA1c, blood cholesterol, medication for blood pressure and exogenous hormone, and medical condition of hypertension. For Models 2 and 3, non-alcoholic dietary food and nutrient variables were additionally adjusted for alcohol intake. Negative associations are depicted in blue, while positive associations are shown in red. The size of the dots in the Manhattan plot represents the strength of the associations. Factors that are significant in both Model 1 and Model 2 will be screened and presented in Model 2. Factors that are significant in all models will be screened and presented in the Model 3 column.

**Figure 3 nutrients-16-00103-f003:**
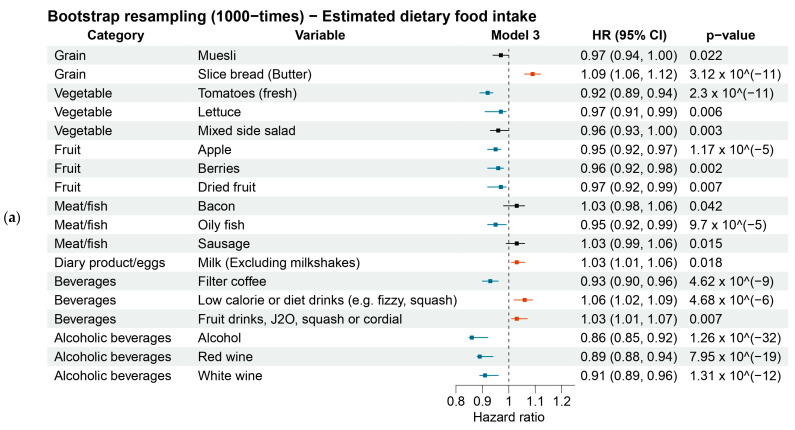
Bootstrap resampling was performed 1000 times to evaluate the association between significant dietary food factors and T2DM risk (**a**) and the association between significant dietary nutrient factors and T2DM risk (**b**). Abbreviation: HR, hazard ratio; CI, confidence interval. The HR represents the increased T2DM risk for each quintile increase in dietary factor consumption. Model 3 is adjusted for gender and age at recruitment, ethnic background, smoking status, physical activity, education level, Townsend index, BMI, HbA1c, blood cholesterol, and medication for blood pressure and exogenous hormone, and medical condition of hypertension. For non-alcoholic dietary food and nutrient variables were additionally adjusted for alcohol intake. Negative associations are depicted in blue; positive associations are shown in red. The 95% CIs were rounded to two decimal places. If the value crosses or is equal to 1.00, it is coloured in black.

**Figure 4 nutrients-16-00103-f004:**
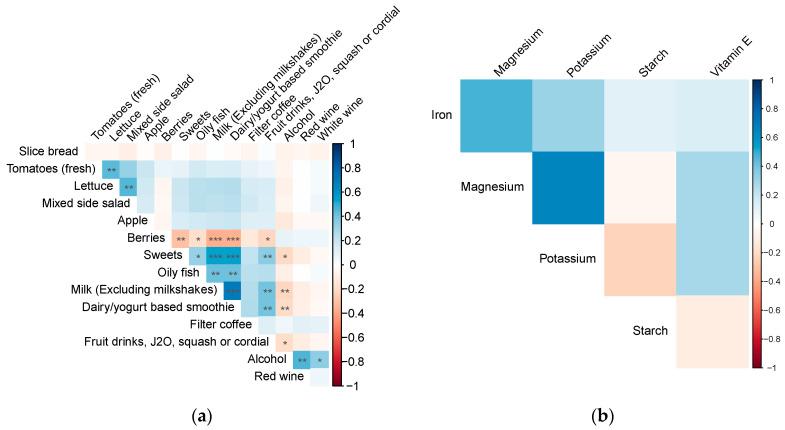
Heatmap showing Spearman’s correlations between significant dietary foods (**a**) and nutrients (**b**) that were identified in the DWAS. Negative correlations are depicted in red, while positive correlations are shown in blue. The intensity of the colours represents the strength of the correlation, with darker shades indicating stronger positive or negative correlations. Symbols ***, **, * represented the correlation between two dietary food factors with *p*-values < 0.001, *p*-value < 0.01 and *p*-value < 0.05, respectively.

**Table 1 nutrients-16-00103-t001:** Baseline characteristics according to the follow-up T2DM status. Abbreviations: T2DM, Type 2 diabetes mellitus; N, sample size; SD, standard deviation; body mass index (BMI). The *p*-value is a comparison based on the X^2^ test and ANOVA analysis for the non-T2DM and the T2DM onset individuals. X^2^ tests were conducted for gender, ethnic background, smoking status, education level, medicine for exogenous hormones, blood pressure, and medical condition of hypertension. ANOVA analysis was conducted for age at recruitment, alcohol intake, physical activity, Townsend index, BMI, Haemoglobin A1C (HbA1c), and blood cholesterol.

	Overall	Non-T2DM	T2DM Onsets	*p*-Value
N	119,040	115,799	3241	
Gender (%)				<0.001
Female	67,364 (56.6)	66,009 (57.0)	1355 (41.8)	
Male	51,676 (43.4)	49,790 (43.0)	1886 (58.2)	
Age at recruitment (mean (SD))	55.97 (7.85)	55.90 (7.85)	58.61 (7.19)	<0.001
Ethnic background (%)				<0.001
White	115,183 (96.8)	112,126 (96.8)	3057 (94.3)	
White and other mixed background	443 (0.4)	429 (0.4)	14 (0.4)	
Asian and Asian mixed background	1664 (1.4)	1579 (1.4)	85 (2.6)	
Other ethnic background	1750 (1.5)	1665 (1.4)	85 (2.6)	
Smoking status (%)				<0.001
Never smoke	68,527 (57.6)	67,076 (57.9)	1451 (44.8)	
Previous smoker	42,201 (35.5)	40,760 (35.2)	1441 (44.5)	
Current smoker	8312 (7.0)	7963 (6.9)	349 (10.8)	
Alcohol intake status (%)				<0.001
Never drink	3339 (2.8)	3183 (2.7)	156 (4.8)	
Previous drinker	3364 (2.8)	3185 (2.8)	179 (5.5)	
Current drinker	112,337 (94.4)	109,431 (94.5)	2906 (89.7)	
Education level (%)				<0.001
Low level	7894 (6.6)	7460 (6.4)	434 (13.4)	
Intermediate level	56,210 (47.2)	55,119 (47.6)	1091 (33.7)	
High level	54,936 (46.1)	53,220 (46.0)	1716 (52.9)	
Physical activity (MET-min/week) (Mean [SD])	2427.62 (2356.48)	2432.42 (2352.23)	2256.11 (2497.81)	<0.001
Townsend Index (mean (SD))	−1.63 (2.84)	−1.65 (2.83)	−1.08 (3.07)	<0.001
BMI (kg/m^2^) (Mean [SD])	26.56 (4.45)	26.44 (4.36)	30.65 (5.67)	<0.001
HbA1c (mmol/L) (Mean [SD])	34.81 (4.25)	34.64 (3.91)	40.98 (8.81)	<0.001
Blood cholesterol (mmol/L) (mean (SD))	5.76 (1.06)	5.76 (1.06)	5.59 (1.18)	<0.001
Medicine for blood pressure (%)	18,745 (15.7)	17,472 (15.1)	1273 (39.3)	<0.001
Medicine for exogenous hormone (%)	7101 (6.0)	6968 (6.0)	133 (4.1)	<0.001
Hypertension (%)	26,164 (22.0)	24,686 (21.3)	1478 (45.6)	<0.001

## Data Availability

Data are contained within the article and Appendix A.

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
