# Peer review of "Diet-Wide Association Study for the Incidence of Type 2 Diabetes Mellitus in Community-Dwelling Adults Using the UK Biobank Data"

_nutrients, 2023, doi:10.3390/nu16010103_

Round 1

Reviewer 1 Report

Comments and Suggestions for Authors

This is an interesting paper worth publishing, but in our view it needs a bit of improvement, as described below:

Lines 141 – 144: “To assess the correlation between pseudo-numerical dietary foods and nutrient intake data with baseline characteristics, we used X 2  tests for categorical covariates and variance tests (ANOVA) for continuous covariates.” This seems very general and a bit vague. Have these inferential tests applied in a univariate manner? What were the groups used in ANOVA? Also the software used for the analysis should be specified.

Lines 161 – 164: it seems unlikely that all analyses were done in base R. Therefore, the authors should also specify (at least the main) packages used for the Cox regression and plot generation.

Lines 184-186: “We found that out of 224 dietary intake factors, 34 were significantly associated with the  onset risk of T2DM, after adjusting p-values in Model 1 using the Bonferroni correction  method.” The authors mention 224 dietary intake factors, but they are nowhere described in the paper. A list with these factors should be shown as a supplementary electronic material.

It is also curious to see baked beans as positively associated with T2DM, as beans are foods with a low glycemic index and tend to be regarded rather as protective. The American Diabetes Association advises people with T2DM to add beans in their diet. “However,  our  study's finding regarding the added risk of baked bean intake on T2DM risk was contrary  to a previous study on legume consumption” Not only this, but it has been shown in a small cross-over clinical trial that “Pinto, dark red kidney and black beans with rice attenuate the glycemic response compared to rice alone.” (https://nutritionj.biomedcentral.com/articles/10.1186/1475-2891-11-23). We support the authors suggestion about the potential confounding factor in this case, but in our view the discussion could be expanded a bit to consider more widely the available evidence.

Lines 403 – 408: It has been shown that “Much of the association between sugars and T2DM is eliminated by adjusting data for body mass index (BMI)”. As the authors in this paper have made adjustments for BMI, indirectly they actually adjusted for sugar consumption, and thus it seems to us that the claimed protective effect of sugar is spurious. This is not very different from other studies that claimed that saturated fats are healthy, because no association is found after adjusting for several co-variates (which are in fact causally associated with saturated fats). In our view, therefore, the authors should question more strongly the association found (which could more likely be the result of confounding).

“Finally, the study's generalizability to other ethnicities may be limited as the  data was obtained from the UK population, with over 97.5% of the data coming from individuals with a White Caucasian background.” Not only this, but the sub-sample is not necessarily representative for the whole population, as indicated by the proportion between males and females, that is different from that in the general population.

The authors apparently have not verified the proportional hazard assumption and its sensitivity. Considering the apparently spurious effects seen for beans and sugar, it would be useful to have some sensitivity analyses where to analyze and test this assumption. They have also not reported on outliers and if any such outliers could have an influence on the results. They should also try cross-validation or bootstrap resampling to assess the stability of the model results.

Author Response

Dear Editor,

Manuscript ID: nutrients-2673200

Manuscript Title: Diet-wide association study for the incidence of type 2 diabetes mellitus in community-dwelling adults using the UK Biobank data

We are most grateful for the reviewers’ valuable and helpful comments, and for the opportunity to have our manuscript reconsidered with additional revisions. We have highlighted the revisions within the manuscript, in the hope that addresses the concerns and issues raised by the editors and reviewers.

Sincerely,

Jiahao Liu

Reviewer 2 Report

Comments and Suggestions for Authors

 I have read and analyzed the manuscript from Liu and coauthors. In my opinion, the present study is very interesting and it can be published in Nutrients in future. However, I have a couple of questions for the discussion and potential manuscript improvements.

  1. The selected patients were between the ages of 40 and 69. What about the younger patient’s cohort? Did the authors obtain results which are relevant only for specific cohort?

  2. Ethnicity quantification. Why did authors evaluate just white or non white? It is well known that Hispanic or Asian patients have very specific features of T2DM development.

  3. Authors used a Tanita body composition analyzer. Why did authors not include in the study visceral fat percentage and other parameters from this analyzer?

  4. Ethnic background and Smoking status parameters in Table 1 have value “Missing”. If the value of these parameters is unknown, I think that these patients should be excluded from the analysis.

Author Response

(The authors gave the same response as above.)

Round 2

Reviewer 1 Report

Comments and Suggestions for Authors

The authors have implemented several of our recommendations and the manuscript has now improved. However, the following have not been addressed, and no justification has been provided:

Lines 141 – 144: “To assess the correlation between pseudo-numerical dietary foods and nutrient intake data with baseline characteristics, we used X 2  tests for categorical covariates and variance tests (ANOVA) for continuous covariates.” This seems very general and a bit vague. Have these inferential tests applied in a univariate manner? What were the groups used in ANOVA? Also the software used for the analysis should be specified. No change has been operated in response to this recommendation, neither has any explanation or rebuttal been provided.

"They should also try cross-validation or bootstrap resampling to assess the stability of the model results." - No report or rebuttal was provided on this recommendation.

Reviewer 2 Report

Comments and Suggestions for Authors

Many thanks to the authors for the comprehensive response. Points 1-2 and 4 - Ok. Point 3 - I asked this question because authors performed the study on type 2 diabetes population. And the question about visceral fat percentage is not the idle question, visceral fat percentage can be closely related with type 2 diabetes incidence. But the сorrections about this point can be maked up to the Editor's opinion.
